# Hot-Spot-Specific Probe (HSSP) for Rapid and Accurate Detection of KRAS Mutations in Colorectal Cancer

**DOI:** 10.3390/bios12080597

**Published:** 2022-08-04

**Authors:** Hyo Joo Lee, Bonhan Koo, Yoon Ok Jang, Huifang Liu, Thuy Nguyen Thi Dao, Seok-Byung Lim, Yong Shin

**Affiliations:** 1Department of Biotechnology, College of Life Science and Biotechnology, Yonsei University, 50 Yonsei-ro, Seodaemun-gu, Seoul 03722, Korea; 2Department of Colorectal Cancer, Asan Medical Center, University of Ulsan College of Medicine, 88 Olympic-ro 43-gil, Songpa-gu, Seoul 05505, Korea

**Keywords:** cancer diagnostics, point mutation, precision medicine, early detection

## Abstract

Detection of oncogene mutations has significance for early diagnosis, customized treatment, treatment progression, and drug resistance monitoring. Here, we introduce a rapid, sensitive, and specific mutation detection assay based on the hot-spot-specific probe (HSSP), with improved clinical utility compared to conventional technologies. We designed HSSP to recognize *KRAS* mutations in the DNA of colorectal cancer tissues (HSSP-G12D (*GGT→GAT*) and HSSP-G13D (*GGC→GAC*)) by integration with real-time PCR. During the PCR analysis, HSSP attaches to the target mutation sequence for interference with the amplification. Then, we determine the mutation detection efficiency by calculating the difference in the cycle threshold (*C*_t_) values between HSSP-G12D and HSSP-G13D. The limit of detection to detect *KRAS* mutations (*G12D* and *G13D*) was 5–10% of the mutant allele in wild-type populations. This is superior to the conventional methods (≥30% mutant allele). In addition, this technology takes a short time (less than 1.5 h), and the cost of one sample is as low as USD 2. We verified clinical utility using 69 tissue samples from colorectal cancer patients. The clinical sensitivity and specificity of the HSSP assay were higher (84% for *G12D* and 92% for *G13D*) compared to the direct sequencing assay (80%). Therefore, HSSP, in combination with real-time PCR, provides a rapid, highly sensitive, specific, and low-cost assay for detecting cancer-related mutations. Compared to the gold standard methods such as NGS, this technique shows the possibility of the field application of rapid mutation detection and may be useful in a variety of applications, such as customized treatment and cancer monitoring.

## 1. Introduction

Colorectal cancer (CRC) is one of the most common cancers in the world. It is ranked third in incidence and second in mortality among human cancers [1,2,3,4]. In 2022, 151,030 new cases and 52,580 deaths were estimated in the United States by the Surveillance, Epidemiology, and End Results (SEER) Program [5]. This reported rate represented an increase of nearly 10% in both all new cancer cases and all cancer deaths. The metastatic disease will develop in approximately 20% of CRC patients diagnosed [6], and it is associated with lower disease-specific survival, with a less than 14% 5-year survival rate [7]. To increase the survival rate, it is helpful to identify the molecular characteristics of tumors and treat them accordingly. Metastatic CRCs are classified based on their oncogene mutations, such as *KRAS*, *BRAF*, and *PIK3CA*. Among these, *KRAS* mutations are discovered in 30–40% of CRCs [8,9]. Ninety percent of the various types of *KRAS* mutations occur in codons 12 and 13, which leads to constitutive activation of epidermal growth factor receptor (EGFR) downstream pathways in CRC cells, resulting in cell proliferation, migration, and metastasis, evasion of apoptosis, or angiogenesis. Both the *G12D* type occurring in codon 12 and the *G13D* type occurring in codon 13 had a high incidence in CRC. The *G12D* type has the highest incidence among *KRAS* mutations, and the *G13D* type is found most frequently in codon 13 [9]. *KRAS* mutations have been investigated for their potential as prognostic biomarkers. *KRAS* wild-type patients showed a therapeutic effects with anti-EGFR monoclonal antibodies (MoAbs) such as cetuximab and panitumumab, whereas patients with *KRAS* mutations had no significant differences [10,11,12,13]. *KRAS* mutations are known to activate the rat sarcoma/mitogen-activated protein kinase (RAS/MAPK) signaling pathway downstream of EGFR without ligand binding to the receptor. Thus, the European Medicines Agency (EMA) and the U.S. Food and Drug Administration (FDA) have recommended the use of cetuximab and panitumumab only in *KRAS* wild-type patients, based on confirmed preclinical and clinical data [11,14,15]. Therefore, detection of *KRAS* mutations is essential both for managing patients with different types of cancer and for providing customized treatment.

Mutant detection technology for the detection and profiling of rare nucleic acid variants with low allele frequencies, such as cancer mutations or resistance, provides patients with the best-customized treatment, early diagnosis, treatment progression, and tumor drug resistance monitoring [16,17,18]. Current methods for detecting mutations include next-generation sequencing and Sanger sequencing, the amplification refractory mutation system, high-resolution melting analysis, dual priming oligonucleotide, allele-specific hydrolysis, and the dual hybridization probe [19]. The gold standard method for DNA sequencing to detect mutant detection is next-generation sequencing, or Sanger sequencing. However, sequencing techniques are complex, time-consuming, and have high costs. Moreover, the complexity of the clinical sample leads to the missing of mutations by direct sequencing methods, as the sample contains only small numbers of mutant cells mixed with an excess of normal cells. To determine the mutant subgroup through direct sequencing, mutant cells must exist at a minimum concentration of at least 30% of the total gene content [19,20]. BEAMing (Bead, Emulsion, Amplification, and Magnetics) techniques and the Idylla method can detect mutations at a limited sensitivity of 1–5% [21]. The application of these methods in clinical oncology, however, is limited. Although these techniques are more accurate and sensitive than sequencing, they need additional equipment and are complex and therefore not suitable for routine clinical samples. To solve these limitations, many novel methods based on PCR that are designed to detect mutations have been studied and reported, such as PCR restriction fragment length polymorphism mapping, allele-specific PCR, and droplet digital PCR. However, these methods also have a limitation in their simultaneous analysis of different types of mutations [22]. Furthermore, recent studies have shown that detection kits from different manufacturers exhibit inconsistent results, even if the same limit of detection (LOD) is used for mutation detection [23]. Conventional methods for detecting the presence and frequency of mutations by quantitative PCR use fluorescent probes specific to wild-type and mutant alleles. This approach is highly sensitive and reliable but detects only one mutation encoded in the probe [24]. In addition, sequence-selective blockers were used to recognize mutations and increase the discrimination efficiency between different types of mutations or wild-types. Substantial research has been focused on mutation detection methods that use sequence-selective blockers to prevent the amplification of wild-type templates [19,25]. However, the technique is still unsatisfactory as a clinical assay. Therefore, a novel technology would be desirable to overcome the limitations of conventional methods.

In this study, we report the clinical utility of a hot-spot-specific probe (HSSP) for the detection of specific mutations. We designed HSSP for testing the detection efficiency of *KRAS* mutations that occur specifically in CRC (HSSP-G12D (*GGT**→GAT*) and HSSP-G13D (*GGC**→GAC*)). We demonstrate the capability of HSSP to discriminate mutant target sequences directly. For the determination of the mutation status, we used both HSSPs and measured the differences in the cycle threshold (*C*_t_) values between HSSP-G12D and HSSP-G13D. HSSP can be easily integrated with conventional methods, such as real-time PCR (qPCR) and direct sequencing, for mutation detection in clinical specimens. Using this assay, the LOD to detect *KRAS* mutations (*G12D* and *G13D*) was 5–10% of the mutant allele in wild-type populations. It is superior to the conventional methods (≥30% mutant allele). Next, 69 tissue samples from CRC patients were used for testing clinical utility. The clinical sensitivity and specificity of the HSSP assay were higher than those of the direct sequencing assay. Therefore, HSSP is a rapid, low-cost, sensitive, and specific assay to detect mutant alleles. It would be possible to provide a fast and correct delivery of patient-customized treatment using this mutation detection assay.

## 2. Materials and Methods

### 2.1. Design of Primer and HSSP

The primer and HSSP sequences are listed in Appendix A. The primer has an amplification product size of 133 bp, suitable for use in qPCR. The mutation is contained within 133 bp. The reverse primers are designed near the point mutations. HSSP overlaps eight nucleotides with the 3′-end of the reverse primer and includes the mutations, so it is designed to attach only to the target mutation. It was also modified with a C3 spacer at the 3′-end of HSSP because it should not be amplified. Adding this modification to the 3′-end can prevent elongation during PCR without significantly affecting the annealing properties [26,27]. Thus, HSSP competitively interferes with the primers to prevent DNA amplification.

### 2.2. HSSP with qPCR

The qPCR process conditions consisted of an initial denaturation step at 95 °C for 15 min, followed by three-step cycling: 45 cycles at 95 °C for 10 s, 63 °C for 20 s, and 72 °C for 20 s, and melting steps at 95 °C for 30 s, 65 °C for 30 s, and 95 °C for 30 s. PCR amplification was performed in a total volume of 20 µL, containing 5 µL DNA, 10 µL of AccuPower 2X GreenStar qPCR Master Mix (Bioneer, Daejeon, Korea), 2.5 µM of each primer, and 25 µM HSSP on a QuantStudio 3 Real-Time PCR System (Applied Biosystems, Waltham, MA, USA), following the manufacturer′s instructions. An optimization experiment was conducted to determine the concentration of HSSP. The optimal concentration was 10 times that of a primer.

### 2.3. Cancer Cell Line

A cell line with *KRAS* mutation was used to confirm the operation and efficiency of HSSP. The *G12D* point mutation was confirmed in a gastric adenocarcinoma cell line (AGS; ATCC CRL-1739), and the *G13D* point mutation was confirmed in a human CRC cell line (HCT116; ATCC CCL-247). The cells were cultured in high-glucose Dulbecco′s modified Eagle′s medium (DMEM, HyClone, Logan, UT, USA), supplemented with 10% fetal bovine serum (FBS, HyClone) and 1% antibiotic-antimycotic (Gibco, Grand Island, NY, USA). The cell culture was maintained in plastic culture dishes and incubated in a humidified atmosphere of 5% CO_2_ at 37 °C. Subcultures were done at a ratio of 1:3 when the cell density reached 80–90% every 3 or 4 days.

### 2.4. Conventional PCR and Sequencing Analysis

The end-point PCR process was comprised of an initial denaturation step at 95 °C for 15 min, followed by 45 cycles at 95 °C for 30 s, 63 °C for 30 s, and 72 °C for 30 s, as well as a final elongation step at 72 °C for 5 min on a T100TM Thermal Cycler (Bio-Rad, Hercules, CA, USA). Five microliters of DNA were amplified in a total volume of 25 µL, containing 10X PCR buffer (Qiagen, Hilden, Germany), 2.5 mM MgCl_2_, 0.25 mM of deoxynucleotide triphosphate, 2.5 µM of each primer, and 1 U of Taq DNA polymerase (Qiagen). PCR products were analyzed by electrophoresis on a 2% agarose gel. The gel was visualized using a ChemiDoc XRS+ System (Bio-Rad). For direct sequencing of DNA, all the DNA samples were amplified and were then purified using the Expin PCR SV (GeneAll, Seoul, Korea). Using the forward primer, the purified samples were directly sequenced on an ABI 3730XL system (Applied Biosystems) at the Macrogen sequencing center (Macrogen, Inc. Seoul, Korea).

### 2.5. DNA Template Preparation

To confirm the LOD and efficiency for HSSP, mutation DNA templates were made using DNA extracted from the AGS and HCT116 cell lines. The DNA was extracted using the QIAamp DNA Mini Kit (Qiagen). In addition, a template was made with DNA obtained from a sample of a patient diagnosed with the wild-type. The DNA template has a length of 777 bp containing *KRAS* mutation. DNA templates were made by PCR analysis using a designed primer and were then purified using the Expin PCR SV (GeneAll). The concentration and quality of the produced DNA templates were assessed using a Qubit 3 Fluorometer (Thermo Fisher Scientific, Waltham, MA, USA), and the DNA copy number was calculated using the EndMemo DNA/RNA copy number calculator. The DNA template was stored at –20 °C until use.

### 2.6. Clinical Sample Collection

To confirm the efficacy in clinical samples, we used a total of 69 cancer samples, including 25 with the *G12D* mutation, 25 with the *G13D* mutation, and 19 with the wild-type. Based on frozen tissue availability, these samples were obtained from the Bio-Resource Center (BRC) of Asan Medical Center (Seoul, Korea) after approval from the Institutional Review Board (IRB No. 2016–0809). The colorectal cancer stage characteristics of all patients are shown in Appendix A. There is no stage I sample (0%). The CRC stages of these samples provided 4 (5.8%) in stage II and 13 (18.8%) in stage III, with the largest number of patients 52 (75.4%) in stage IV [28]. The genomic DNAs from the tissues were extracted using ATL buffer with proteinase K from a QIAamp DNA Mini Kit (Qiagen) according to the manufacturer′s instructions. The samples were eluted using 100 µL of elution buffer. The eluted DNA was stored at –20 °C until use.

## 3. Results

### 3.1. Principles of HSSP

The operating principles of HSSP are illustrated in Figure 1. HSSP overlaps the target primer sequence, includes a single mutation, and is modified with a C3 spacer at the 3′-end to prevent amplification by qPCR. When a mutant sequence exists, the HSSP competitively attaches to the target mutant sequence, thereby preventing the primer from binding to the target mutant sequence; subsequently, the sequence is not amplified. We designed HSSP for testing the detection efficiency of *KRAS* mutations that occur specifically in CRC, namely, *G12D* (*GGT**→GAT*) and *G13D* (*GGC**→GAC*). In the case of wild-type with no *KRAS* mutations, the *KRAS* sequences were amplified by the primers; the HSSPs (HSSP-G12D and HSSP-G13D) could not block the amplification. If the *G12D* mutation exists, HSSP-G12D blocks the target amplification. However, HSSP-G13D does not block the amplification in the *G12D* target DNAs. Conversely, if the *G13D* mutation exists, HSSP-G13D blocks the target amplification. However, HSSP-G12D does not block the amplification in the *G13D* target DNAs (Figure 1-upper). For the determination of the mutation status, we measured the differences in the *C_t_* values between HSSP-G12D and HSSP-G13D. In the wild-type sequence, the difference (Δ*C_t_*) was 2 to −2 based on the equation:$$\Delta C_{t}=C_{t}\left(\text{HSSP-G12D}\right)-C_{t}\left(\text{HSSP-G13D}\right)$$

When the *G12D* mutation exists, Δ*C_t_* is greater than 2. In the case of *G13D* mutation, Δ*C_t_* is less than −2. This method is a simple and easy approach that allows the distinguishing of mutations within 1.5 h and is a rapid and robust assay compared to the commonly used qPCR and sequencing methods.

### 3.2. Optimization and Application of HSSP

Prior to the use of HSSP, the efficiency of the target primer was confirmed. The LOD of the primers was tested with either the *G12D* mutant DNA obtained from the AGS cell line, the *G13D* mutant DNA obtained from the HCT116 cell line, or the wild-type DNA without mutations. It was confirmed that the LOD of the primers was 1 × 10^2^ copies of the DNA from each of the three types (Figure 2A). In addition, we used DNA templates from the cell lines AGS and HCT116 to confirm the limit of detection with HSSP. The DNAs were obtained by serial dilution of cell concentrations from the two cell lines. Detection was confirmed with sample sizes down to 10 cells when the test was performed under the same conditions with cells from the two cell lines (Appendix A). Therefore, by using HSSP, the mutations can be detected and classified even in small numbers of cells or by DNA.

Next, to find out the optimal concentration of HSSP, we used the serially diluted HSSP-G12D or HSSP-G13D (5–50 µM) for detection of the *KRAS* 12 or 13 mutation. The Δ*C_t_* value of qPCR was larger in 50 µM HSSP (Figure 2B) than in other concentrations, indicating that higher HSSP concentration could block the target DNA amplification by interfering with the binding of the primer to the target sequence. Although the *C_t_* value with 50 µM HSSP was higher than that with 25 µM HSSP, the difference was not significant. Therefore, we selected 25 µM HSSP for HSSP-G12D (Figure 2B-left) and HSSP-G13D (Figure 2B-right) with good stability, efficiency, and low cost.

After the optimization, we conducted the direct sequencing and gel electrophoresis assays to confirm the utility of HSSP as a mutant blocker (Figure 2C–E). We used the direct sequencing assay to confirm heterozygous *G12D* and *G13D* mutations in the AGS and HCT116 cell lines, respectively. Working with HSSP-G12D in the AGS cell line, the *G12D* mutant signal (*GGT**→GAT*) was significantly reduced. Conversely, working with HSSP-G13D, the *G12D* mutant signal was enhanced by reducing the wild-type signal (Figure 2C). In the case of the *G13D* type mutation in the HCT116 cell line, the *G13D* mutant signal (*GGC→GAC*) was significantly reduced when HSSP-G13D was used. However, the *G13D* mutant signal was enhanced by reducing the wild-type signal when HSSP-G12D was used (Figure 2D). Therefore, HSSP reduces the signal of the target mutation, and the wild-type signal is also reduced, so that the mutation signal is comparatively enhanced. The results from direct sequencing were similar to the results of the gel electrophoresis assay (Figure 2E). When we used HSSP-G12D and HSSP-G13D in the wild-type target, both HSSPs interfered the target amplification. In the case of *G12D* and *G13D* cell lines, HSSP-G12D and HSSP-G13D interfered with the target mutant amplification only (Figure 2E). Taken together, the HSSPs specifically blocked the target mutant amplification in cellular applications.

### 3.3. Detection Limit of HSSP in Mixed-Cell Populations

To check whether HSSP can distinguish the single-point mutation (*G12D* or *G13D*) in mixed-cell populations, we used serially diluted DNA samples from both AGS and HCT116 cell lines. Genomic DNA was obtained from the cell mixtures containing 0–100% of the mutant DNAs in the wild-type DNA. When HSSP-G12D and HSSP-G13D were used to amplify the mutations using qPCR, the mutant target could be amplified in the mixture containing 10% of both *G12D* (Figure 3A) and *G13D* (Figure 3B). A previous report confirmed that PCR and direct sequencing could not detect the mutant allele in the mixed population containing >30% mutant templates [28]. In contrast, the qPCR assay with HSSP could detect the mutant allele in the mixed samples containing 10% mutant templates in mixed sample populations. For accurate detection, we set an accurate standard for the Δ*C_t_* value of qPCR. When the level of the mutant template was 10%, the average difference value was 1.8 in *G12D* and −2.4 in *G13D*. In addition, using a wild-type DNA template, we checked from 10^5^ to 10 copies and found that the Δ*C_t_* values were 0.662, 1.29, 1.43, 0.84, and 0.47, respectively. The maximum value was about 1.5, and the average difference value was about 0.98 (Appendix A). As a result, we have confirmed that the wild-type is included within the sequence range based on the case with a 10% mutant template. Thus, the Δ*C_t_* value range of the mutation detection criteria was determined at between 2 and −2.

In the case of serial dilution, when the level of the mutant template was 5%, the same pattern was shown as the wild-type. For more sensitive detection of the mutant DNAs, we conducted a direct sequencing assay with HSSP. When direct sequencing was used to detect the *G12D* and *G13D* mutations, the mutant sequences could be identified at a 1% mutant template using HSSP (Appendix A). The wild-type strand cannot be amplified; only the mutation strand is emphasized. This confirmed that HSSP could detect nearly 1% of mutations using direct sequencing with both HSSP-G12D (Appendix A) and HSSP-G13D (Appendix A).

### 3.4. Clinical Utility of HSSP

To validate clinical utility, we randomly selected 69 frozen tissue samples from patients with CRC treated at the BRC of Asan Medical Center. Among the 69 samples, 25 with the *G12D* mutation (36.2%), 25 with the *G13D* mutation (36.2%), and 19 with no mutation (27.6%) were examined at the BRC and were used as a reference [28]. HSSP with qPCR results in 69 clinical samples showed differences in *C_t_* values (Figure 4A). After applying the Δ*C_t_* range (−2, 2), most wild-type patients had a value between 2 and −2, most G12D type patients had a value of more than 2, and most G13D type patients showed a value of less than −2. For comparison with the conventional method, the direct sequencing method was also performed using the same samples. From the results of HSSP with qPCR, twenty-one samples were detected as true positives (TP) and four as false negatives (FN), out of twenty-five samples with the *G12D* mutation. Among the twenty-five samples with the *G13D* mutation, twenty-three were detected as TP, and two as FN. In the case of the nineteen wild-type samples, seventeen were detected as TP and two as FN. With the direct sequencing assay, twenty samples were detected as TP and five as false positives (FP), out of twenty-five samples with the *G12D* mutation. In twenty-five samples with the *G13D* mutation, twenty were detected as TP and five as FN. In the case of the wild-type samples, nineteen out of nineteen patients were detected as TP. Detailed data for each patient′s results can be confirmed in Appendix A. Based on these results, the clinical specificity and sensitivity were calculated (Table 1). When we compared the clinical specificity and sensitivity of the two methods, HSSP with the qPCR assay showed higher sensitivity: 84% and 92% for *G12D* and *G13D* mutations, respectively. The specificity of HSSP with the qPCR assay was comparable to that of the direct sequencing assay. Looking at these results, it was possible to see the high sensitivity and specificity of HSSP with the qPCR assay for mutation detection. Therefore, the HSSP assay showed more accuracy in diagnosing mutations. By accelerating the diagnosis, the HSSP assay is expected to allow faster and more appropriate treatment for patients with cancer.

## 4. Discussion

Mutant detection is important for the potential clinical prognostic in cancer monitoring, ranging from customized treatment to tumor drug resistance monitoring. Various technologies to detect single-point mutation have been developed and modified for rapid and highly sensitive detection for clinical use. In this study, we present a rapid and simple technique for mutant detection using a novel probe, HSSP. The presented technology shows a good efficiency compared to existing widely used methods, such as real-time PCR and direct sequencing (Table 2). Other methods are relatively time-consuming, expensive, or reliant on specialized equipment. The HSSP with qPCR test can detect the mutation rapidly (within 1.5 h). This takes less time than does the direct sequencing assay, which takes at least several days to deliver the result, and is faster than other PCR-based methods. In addition, in terms of cost, it is less than USD 2 per sample, which is cheaper than other methods. Differences in *C_t_* values can easily be checked using this analysis technique, so it can be easily used to detect other mutations, and this helps with rapid analysis. The LOD of HSSP (5–10% of mutant allele) is higher than those of other assays, such as qPCR and direct sequencing (≥30% of mutant allele). Hence, HSSP with qPCR is highly useful for the detection of hot spot mutations in clinical specimens, more so than any sequencing assay.

In this study, 69 tissue samples were used to detect *KRAS* mutation. As a result, the clinical sensitivity of HSSP with the qPCR assay was shown to be superior to that of the direct sequencing assay (Table 1). In particular, the detection of wild-type showed higher specificity than in the direct sequencing assay. The HSSP with qPCR method will improve efficiency, especially when treating *KRAS* wild-type, including those requiring anti-EGFR MoAbs. Therefore, these results show significant potential for rapid and accurate mutation detection in clinical samples. In addition, this HSSP mutation detection assay will be of great help to patient-customized treatment.

Despite the advantages of this technique, the limitations of this study should be addressed in future studies. First, HSSP can be easily applied to qPCR. However, the latter was performed twice for the detection of a single mutation to compare the differences in *C_t_* values from two other mutations. The HSSP method can be difficult to perform with a large number of samples at once. Second, the technique is a negative selection assay, one which blocks the target mutation by HSSP. For example, HSSP-G12D binds with the mutant allele to interfere with the amplification, and other wild-type alleles could then be amplified with HSSP-G13D. Based on the *C_t_* values, the mutation can be determined. Therefore, both target and non-target HSSP must be used for the detection, and there is a limitation in the fact that it is difficult to detect mutations other than those designed. Third, the clinical samples are not enough to validate this technique. To overcome this limitation, further study would be desired with a larger cohort of clinical samples, varying by cancer stages as well as other samples, such as tissue and liquid biopsy, for testing the versatility of the HSSP. Finally, this HSSP assay should be applied to sample preparation techniques for clinical diagnostic systems [29,30,31]. Overall, this method is expected to be able to detect not only *KRAS* mutations but also other mutations by selectively designing a different sequence. We envision that HSSP will be useful in a variety of applications, such as customized treatment and cancer monitoring.

## Figures and Tables

**Figure 1 biosensors-12-00597-f001:**
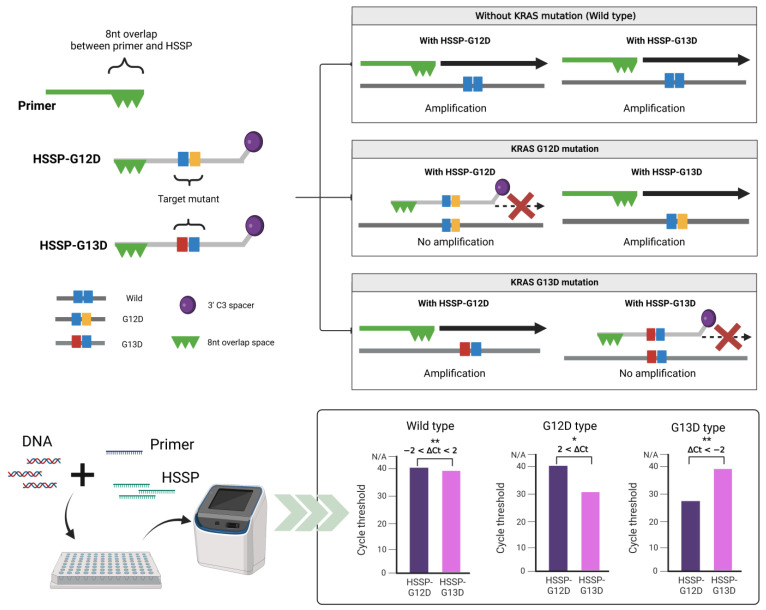
Schematic representation of the principle of HSSP. The hot spot-specific probe (HSSP) is designed to overlap with PCR primers and contains a specific sequence of single mutations. DNA of the target mutation sequence competitively attaches to the HSSP before the primer binds, which prevents amplification. In the case of sequences different from HSSP, the primer is attached and amplified. In the case of HSSP with qPCR, mutants can be checked by detecting a mutation from the difference in the *C*_t_ values within 1.5 h. Calculation of the difference between the *C_t_* values when using HSSP-G12D and HSSP-G13D. If it is higher than 2, it has the *G12D* mutant; if it is lower than −2, it has the *G13D* mutant. A value between 2 and −2 is considered wild-type; * *p* < 0.05, ** *p* < 0.01.

**Figure 2 biosensors-12-00597-f002:**
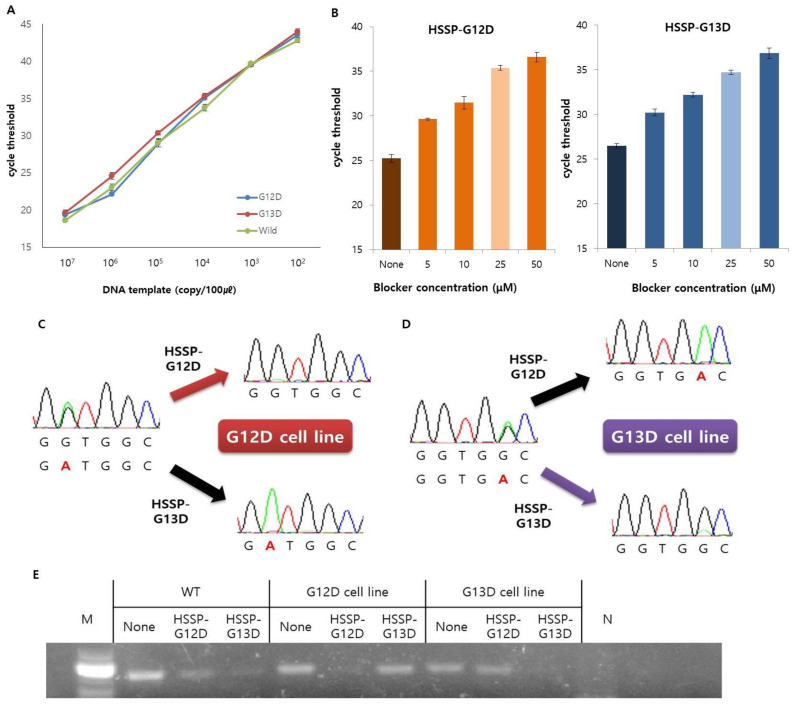
Optimization and application of the *KRAS* mutations primer and HSSP. (**A**) Serial dilution of DNA template was used to test the detection limit of the PCR primer. In the same way, up to 10^2^ copies can be detected in three types. Data are presented as mean ± SD, based on at least three independent experiments. (**B**) Efficiency of the HSSP depends on the concentration. It showed the largest difference at the highest concentration and the most stable value at 25 µM. This is marked in light colors. Controls without HSSP are displayed in deeper colors. Data are presented as mean ± SD, based on at least three independent experiments. (**C**–**E**) Operation of the HSSP was confirmed from direct sequencing and electrophoresis after end-point PCR. This shows that mutations were not amplified in DNA with the same sequence as the HSSP. (**C**,**D**) These render direct sequencing results to confirm the operation of the HSSP on (**C**) *G12D* cell line and (**D**) *G13D* cell line. (**E**) Electrophoresis results after end-point PCR confirmed the operation of the HSSP.

**Figure 3 biosensors-12-00597-f003:**
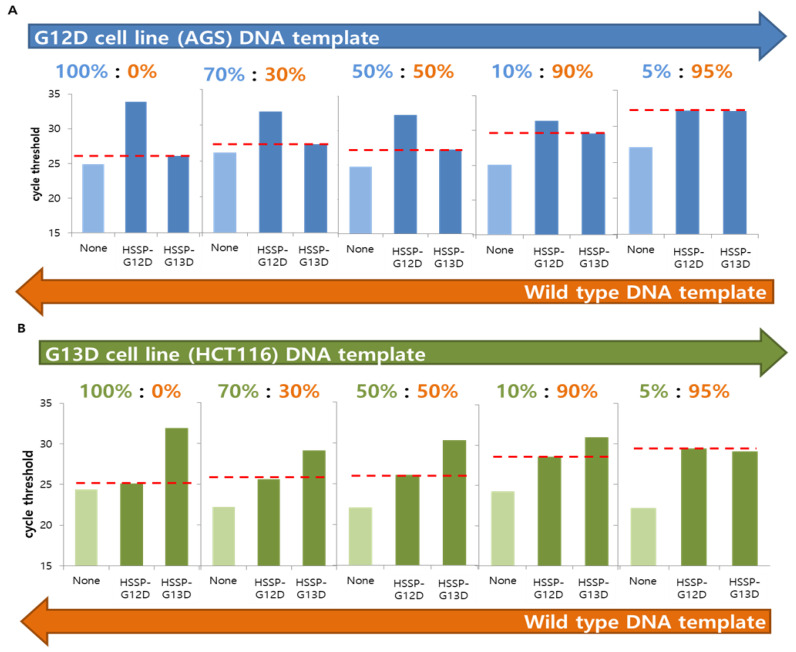
Analysis of the *KRAS* mutations, with a serially diluted mixed DNA template, using HSSP with qPCR. (**A**) *G12D* cell line (AGS) DNA template was mixed by serial dilution with wild-type DNA template, and the detection limit of the mutant was checked. (**B**) *G13D* cell line (HCT116) DNA template was mixed by serial dilution with wild-type DNA template, and the detection limit of the mutant was checked. In both cases, the value was indicated by a red line based on the case of using the non-target HSSP; the *C_t_* value using HSSP-G13D in *G12D* mutation and the *C_t_* value using HSSP-G12D in *G13D* mutation offer visual confirmation of the difference in *C_t_* values. Concentrations of 10% and above produce a difference in the *C_t_* value, indicating that the mutations can be detected using HSSP with qPCR.

**Figure 4 biosensors-12-00597-f004:**
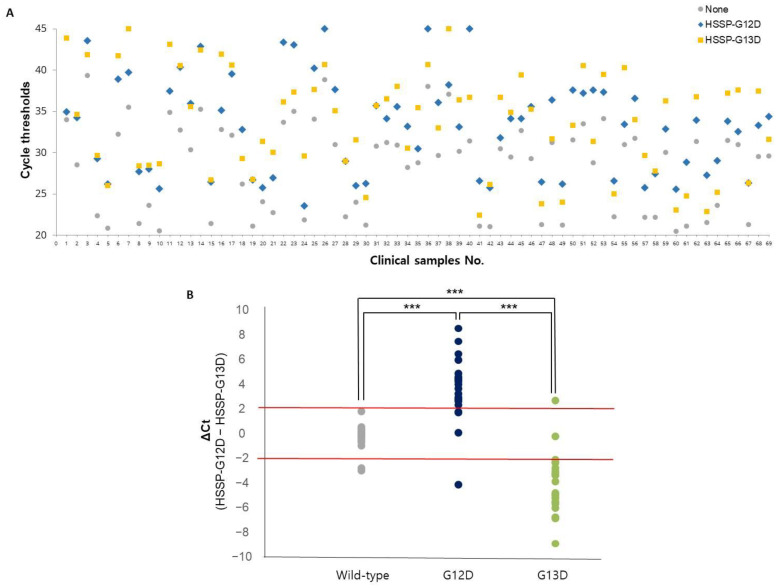
Application of the *KRAS* mutations detection in 69 clinical samples. (**A**) *C_t_* values using HSSP with qPCR in 69 clinical samples. The colors represent HSSP-G12D (blue), HSSP-G13D (yellow), and without HSSP (gray). (**B**) Mutation detection using Δ*C_t_* range (2, −2). Based on the type of sample by BRC, wild-type (gray), G12D type (deep blue), and G13D type (green). The standard for the mutation classification is Δ*C_t_* range (2, −2). Significant differences between groups are indicated; *** *p* < 0.001.

**Table 1 biosensors-12-00597-t001:** Comparison of specificity and sensitivity of HSSP and direct sequencing for *KRAS* mutation detection in clinical samples.

KRAS	HSSP with Real-Time PCR		Direct Sequencing	
	Sensitivity (95% CI)	Specificity (95% CI)	Sensitivity (95% CI)	Specificity (95% CI)
*G12D*	84% (63.92–95.46%)	97.73% (87.98–99.94%)	80.00% (59.30–93.17%)	100% (91.96–100.00%)
*G13D*	92% (73.97–99.02%)	93.18% (81.34–98.57%)	80.00% (59.30–93.17%)	97.06% (87.98–99.94%)
WT	89.47% (66.86–98.7%)	92% (80.77–98.70%)	100% (82.35–100.00%)	82.00% (68.56–91.42%)

**Table 2 biosensors-12-00597-t002:** Comparison of mutation detection technologies for *KRAS* mutation.

	Mutation LOD (%)	Protocol Time (h)	Cost Per Sample (USD)	Refs
HSSP with real-time PCR	5–10	1.5	≤2	This paper
HSSP with direct sequencing	1–2	48+	≤6	
NGS	1	48+	1000+	[32]
AS-PCR	1–10	2	≤3	[24]
Droplet digital PCR	0.1	4	10	[23,33]
OncoBEAM	0.02	48+	24	[34]
Idylla	1	2.5	10	[21]

## Data Availability

The data that support the findings of this study are available from the corresponding author upon reasonable request.

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
