# Peer review of "Hot-Spot-Specific Probe (HSSP) for Rapid and Accurate Detection of KRAS Mutations in Colorectal Cancer"

_biosensors, 2022, doi:10.3390/bios12080597_

Round 1
Reviewer 1 Report
In this article, a specific KRAS gene probe for detecting of colon cancer was developed. HSSP shows the advantages of fast, low-cost and sensitive. However, the author needs to think about the following issues:
1. Can ΔCT range (-2, 2) be applied to the clinical samples used in this article?
2. How to prove that G12D mutation signal is enhanced by reducing wild-type signal? 3. It is suggested to increase the efficiency comparison with the existing detection methods or means that can detect the gene mutation at this site, so as to highlight the advantages of HSSP.
4. Some sentences with semantic repetition are found in the writing, so it is suggested to delete them.
Author Response
We sincerely appreciate the Reviewers for their thoughtful review of our manuscript. We have taken their comments into careful consideration in preparing our revision, significantly improving the quality of the manuscript in the process. Below, we present point-by-point responses to the reviewers’ comments.
Reviewer #1: In this article, a specific KRAS gene probe for detecting of colon cancer was developed. HSSP shows the advantages of fast, low-cost and sensitive. However, the author needs to think about the following issues:
We sincerely appreciate the reviewer for very carefully reading the manuscript, the constructive comments, suggestions, which have helped us significantly improve the quality of the manuscript.
- Can ΔCT range (-2, 2) be applied to the clinical samples used in this article?
Response: We thank the reviewer for this comment. The difference in Ct values when using HSSP with qPCR was measured in the mixed-cell populations and wild-type DNA to determine the standard for classifying mutations. In the mixed-cell population, we confirmed the Ct values at various mutation rates from 100% to 5%. The results showed a clear difference at a 10% rate but did not show the difference at 5%. At a mutation rate of 10%, the ΔCt values are about an absolute value of 2, so we determine the standard for mutation detection as ΔCt values less or greater than the absolute value of 2. In addition, we confirmed the Ct values using wild-type DNA ranging from 105 to 10 copies per reaction in 69 clinical samples. The maximum ΔCt value was about 1.5, and the average ΔCt value was about 1. We confirmed that the ΔCt value of the wild-type target is included within the range of absolute value of 2 for mutation detection. Based on the results, the ΔCt range (−2, 2) was determined as the standard for detection of wild-type and mutation. Among the 25 G12D type samples, 21 were ΔCt values with more than 2, and 23 were ΔCt values with less than −2 among 25 G13D type samples. 17 of 19 wild-type samples show the ΔCt value of between 2 and −2. We have added the detailed results on Figure 4 and page 9 in the revised manuscript.
Figure 4
Figure 4. Application of the KRAS mutations detection in 69 clinical samples. (A) Ct values using HSSP with qPCR in 69 clinical samples. The colors represent HSSP-G12D (blue), HSSP-G13D (yellow), and without HSSP (gray). (B) Mutation detection using DCt range (2, −2). Based on the type of sample by BRC, wild-type (gray), G12D type (deep blue), and G13D type (green). The standard for the mutation classification is DCt range (2, −2). Significant differences between groups are indicated: ***p < 0.001.
- How to prove that G12D mutation signal is enhanced by reducing wild-type signal?
Response: We thank the reviewer for this comment. The enhancement of the G12D mutant signal by reducing the wild-type signal was confirmed by direct sequencing. The results are represented in Figures 2C and D. These results show that DNA amplification efficiency is reduced in mutations with the same sequence of HSSP. The results in G12D cell line showed in Figure 2C. The signal of the G12D mutation (GATGGC) was reduced when HSSP-G12D was used. In addition, the G12D mutation signal was enhanced when HSSP-G13D was used because the wild-type (GGTGGC) signal was also reduced. These results show the same results in the G13D cell line in Figure 2D. When HSSP-G12D was used, the wild-type (GGTGGC) signal is reduced and enhanced the signal of G13D mutation (GGTGAC). It seems that HSSP affects the amplification of the wild-type, as the inclination of the wild-type signal to reduce is one base difference between the target mutation and the wild-type. Moreover, it is confirmed that wild-type is affected, but the Ct value is not significantly different from the case without HSSP in the case of 100% mutation rate in Figure 3. Therefore, HSSP reduces the signal of the target mutation, as well as the wild-type signal is also reduced, so that the mutation signal was comparatively enhanced.
On page 6, Therefore, HSSP reduces the signal of the target mutation, and the wild-type signal is also reduced, so that the mutation signal was comparatively enhanced.
Figure 2 C–D
- It is suggested to increase the efficiency comparison with the existing detection methods or means that can detect the gene mutation at this site, so as to highlight the advantages of HSSP.
Response: We thank the reviewer for this comment. We have revised the manuscript and added advantages of HSSP on page 10 (discussion part) in the revised manuscript.
- Some sentences with semantic repetition are found in the writing, so it is suggested to delete them.
Response: We thank the reviewer for this comment. We have revised the manuscript to remove semantic repetition in the revised manuscript.

Reviewer 2 Report
Please see PDF file attached.

Author Response
We sincerely appreciate the Reviewers for their thoughtful review of our manuscript. We have taken their comments into careful consideration in preparing our revision, significantly improving the quality of the manuscript in the process. Below, we present point-by-point responses to the reviewers’ comments.
Reviewer #2: In summary, the authors developed, optimized and validated a KRAS mutation detection assay based on the hot spot-specific probe (HSSP) and real-time PCR, as well as compared the specificity and sensitivity of their assay with direct sequencing in 69 tissue samples from colorectal cancer (CRC) patients. According to the authors, their assay has decreased sample costs, protocol time, the limit of detection (5-10%) compared to conventional clinically compatible methods (> 30% mutant allele). Their optimization and validation of HSSP could potentially have long-term impact on treatment plans of numerous CRC patients. The authors' work is scientifically relevant and sound, and could be of interest to the research community. However, the quality of presentation of their work is compromised by information missing from their methodology and a clear discussion on the limitations of their HSSP assays. The English language is good for the most part, but there are still concerns with sentences where pronouns are mentioned with no noun to refer to, which can make readers confused.
We sincerely appreciate the reviewer for very carefully reading the manuscript, the constructive comments, suggestions, which have helped us significantly improve the quality of the manuscript.
In the introduction section, the authors clearly describe the relevance of detecting KRAS mutations and the shortcomings of current methods, which are either not sufficiently accurate or not suitable for analysis of clinical samples. However, the authors not only omit clinically relevant issues of application of screening technologies in the clinic, but also fail to discuss how their proposed HSSP + qPCR assay will address such issues and the shortcomings they have mentioned. Furthermore, the limitations of the HSSP + qPCR assay are not discussed in the manuscript. From a clinical perspective, this is a manuscript which attempts to make a technology more clinically useful without discussing what is clinically relevant for the application in question (despite the manuscript being scientific significant - please not scientific relevance is totally different from clinical relevance). I suggest the authors to add the omitted/missing information discussed above.
First, the introduction section does not mention that part of the problem of invasive sample collection for analysis. I do not think the HSSP + qPCR assay developed by the authors could be used in cases where cancer cells cannot be collected. There is no description of how sensitive this assay is for CRC of different stages. There is no reference to how many cells are needed to achieve the specificity and sensitivity values reported later in the text (e.g., Table 2).
Response: We thank the reviewer for this comment. We have confirmed the limit of detection of the HSSP + qPCR assay. The results showed in Figure 2A that the limit of detection is 100 copies/100 μL (5 copies/reaction) using 777bp DNA template. The results were confirmed with each type; G12D, G13D, and wild-type. In addition, to check the minimum number of cells to be detected, DNAs were used by serial dilution of cell concentrations from the AGS and HCT116 cell lines. The limit of detection was confirmed using qPCR. As a result, it was confirmed to detect down to 10 cells when the test was performed under the same conditions from two cell lines. Thus, it can be detected even with a few numbers of cells, so it is possible to detect and classify the mutations by using HSSP. The limit of detection with the cell lines was added on Supplementary Figure S1 in the revised manuscript.
Supplementary Figure S1. Detection limit of the PCR primer in the cell line. Serial dilution of cell concentration was used to test the detection limit of the PCR primer from cell line extracted DNA. In the same way, up to 10 cells/ 100 µL can be detected in AGS cell line (purple) and HCT116 cell line (orange). Data are presented as mean ± SD, based on at least three independent experiments
On page 5–6, In addition, we used DNA templates from the cell lines (AGS and HCT116) to confirm the limit of detection with HSSP. The DNAs were obtained by serial dilution of cell concentrations from two cell lines. It was confirmed to detect down to 10 cells when the test was performed under the same conditions from two cell lines (Supplementary Figure S1). As a result, the mutations can be detected and classified even in small numbers of cells or DNA by using HSSP.
Second, the materials and methods section, especially 2.6. Clinical Sample Collection, does not specify the CRC stages of frozen tissue samples analyzed. Because of the absence of information on CRC stages in both introduction and material and methods sections, it is difficult to evaluate whether the results achieved by the authors are simply due to analyzing cancer in more advanced stages compared to previous studies.
Response: We thank the reviewer for this comment. We used a total of 69 clinical samples from colorectal cancers. The samples were included of 25 G12D types, 25 G13D types, and 19 wild-types without KRAS mutations. The CRC stages of these samples provided 4 (5.8%) in stage Ⅱ and 13 (18.8%) in stage Ⅲ, with the largest number of patients 52 (75.4%) in stage Ⅳ. We did not have a stage Ⅰ sample, so the results of stage Ⅰ could not be confirmed. It is difficult to analyze whether the CRC stage affects the HSSP assay at each stage, so the results for the clinical utility were analyzed without disease stage classification. Therefore, we will use more clinical samples in future studies to validate the clinical utility of the HSSP assay and analyze the differences between CRC stages. The detailed CRC stage of clinical samples was showed on Supplementary Table S2 in the revised manuscript.
Supplementary Table S2. Cancer stage characteristics of CRC patients
|
Cancer stage |
Total No. (%) |
KRAS wild type No. (%) |
KRAS mutation |
|
|
G12D type No. (%) |
G13D type No. (%) |
|||
|
Ⅰ |
0 (0%) |
0 (0%) |
0 (0%) |
0 (0%) |
|
Ⅱ |
4 (5.8%) |
3 (15.8%) |
2 (8%) |
0 (0%) |
|
Ⅲ |
13 (18.8%) |
6 (31.6%) |
2 (8%) |
5 (20%) |
|
Ⅳ |
52 (75.4%) |
10 (52.6%) |
21 (84%) |
20 (80%) |
|
Total no. of patients |
69 |
19 |
25 |
25 |
On page 4, The colorectal cancer stage characteristics of all patients are shown in Supplementary Table S2. There is no stage Ⅰ sample (0%). The CRC stages of these samples provided 4 (5.8%) in stage Ⅱ and 13 (18.8%) in stage Ⅲ, with the largest number of patients 52 (75.4%) in stage Ⅳ.
On page 11, To overcome these limitations, further study would be desired with a larger cohort of clinical samples depend on cancer stages as well as other samples, such as tissue and liquid biopsy.
Third, since the introduction did not mention the number of cells necessary to achieve enough
specificity and sensitivity, it is hard to evaluate whether the authors can even use liquid biopsy or extend their approach to other applications, following their statement: "[ ... ] the clinical samples are not enough to validate this technique. To overcome these limitations, further study would be desired with a larger cohort of clinical samples as well as other samples, such as liquid biopsy."
Response: We thank the reviewer for this comment. We have established the limit of detection by using DNA templates obtained from the cell lines. As a result, the limit of detection was confirmed to be 5 DNA copies/reaction and 10 cells. The HSSP + qPCR assay can detect mutations even with very few numbers of cells or DNA. In this study, HSSP assay was performed using cell line samples and clinical tissue samples. Then, DNA was extracted using a commercial DNA extraction kit (QIAamp DNA Mini Kit) according to the manufacturer's instructions. The results showed high efficiency using extracted DNA from liquid cell line samples and solid tissue samples. HSSP assay can be used by extracting DNA regardless of liquid or solid state. Therefore, we expect that the HSSP assay to be applied to a variety of clinical specimens, including tissue and liquid specimens. In future studies, we will validate the HSSP assay for CRC mutation detection using liquid biopsy. We have added the information on page 5 and 11 in the revised manuscript.
On page 5–6, In addition, we used DNA templates from the cell lines (AGS and HCT116) to confirm the limit of detection with HSSP. The DNAs were obtained by serial dilution of cell concentrations from two cell lines. It was confirmed to detect down to 10 cells when the test was performed under the same conditions from two cell lines (Supplementary Figure S1). As a result, the mutations can be detected and classified even in small numbers of cells or DNA by using HSSP.
On page 11, To overcome these limitations, further study would be desired with a larger cohort of clinical samples depend on cancer stages as well as other samples, such as tissue and liquid biopsy.
A minor revision would be changing figure 4 to represent data of all patients. Here the authors can use statistical tests to check for significant statistical differences between samples of 2 or 3 groups.
Response: We thank the reviewer for this comment. We have modified Figure 4. Significant differences between the groups are indicated as less than 0.001.
Figure 4. Application of the KRAS mutations detection in 69 clinical samples. (A) Ct values using HSSP with qPCR in 69 clinical samples. The colors represent HSSP-G12D (blue), HSSP-G13D (yellow), and without HSSP (gray). (B) Mutation detection using DCt range (2, −2). Based on the type of sample by BRC, wild-type (gray), G12D type (deep blue), and G13D type (green). The standard for the mutation classification is DCt range (2, −2). Significant differences between groups are indicated: ***p < 0.001.
Finally, please revise the manuscript for the English language. I provide some of the questions readers may have from a few parts of the text as an example, but what I show below may not be all occurrences which need to be corrected.
Occurrence 1: What is limited? The authors should make clear in the manuscript text (not to the reviewers only) what "it" refers to.
"It is limited to simultaneous 81 analysis of different types of mutations [22)."
Response: We thank the reviewer for this comment. We have revised the sentence in the revised manuscript.
On page 2, However, these methods also have a limitation to simultaneous analysis of different types of mutations [22].
Occurrence 2: Which method? Using HSSP + qPCR? It is not clear from this and previous sentences as there is nothing "the method" refers to. In particular, the method will improve efficiency, especially when treating KRAS wild-type, such as those requiring anti -EGFR MoAbs.
Response: We thank the reviewer for this comment. We have revised the sentence in the revised manuscript.
On page 10, In particular, the detection of wild-type showed higher specificity than direct sequencing assay. The HSSP with qPCR method will improve efficiency, especially when treating KRAS wild-type, such as those requiring anti-EGFR MoAbs.

Round 2
Reviewer 2 Report
After thoroughly and carefully checking all the authors modifications in the manuscript text, I am convinced that they have addressed all my concerns. The manuscript now looks scientifically sound and relevant, while recognizing clinically important factors as well as limitations of their study. I recommend proceeding with the publication of the authors' manuscript.